# KATANIN promotes cell elongation and division to generate proper cell numbers in maize organs

Stephanie E. Martinez[1], Kin H. Lau [2,6], Lindy A. Allsman[1], Cassandra Irahola[1], Cassetty Habib[1], Isabel Y. Diaz[1], Ian Ceballos[1], Emmanuel Panteris[3], Peter Bommert[4], Amanda Wright[5], Clifford Weil[2] & Carolyn G. Rasmussen [1] ✉

Microtubule severing is essential for proper eukaryotic cell elongation and division. Here we show that the microtubule severing protein, KATANIN p60, is encoded by two genes in *Zea mays* L. (maize) called *Discordia3a (Dcd3a)* and *Dcd3b*. A semi-dominant mutant with short stature, poor fertility, and a clumped tassel was identified in *Dcd3b* called *Clumped tassel1* (*Clt1*). Genetic enhancers that further reduced stature and fertility were identified in inbred lines and mapped to the *Dcd3a* locus, identifying several *dcd3a* alleles. Loss-of-function p60 allele combinations reduce microtubule severing, fertility, and cell elongation. Cell elongation defects, in turn, contribute to G1 delay. KATANIN p60 is important for preprophase band (PPB) formation and positioning, and nuclear positioning in symmetric cell divisions. Misoriented PPBs lead to offset nuclei and rare misoriented symmetric divisions in mutants. A combination of these defects contributes to generating small mutant plants with fewer cells.

Dynamic microtubule reorganization is critical for development in eukaryotes. Microtubule dynamics are regulated by hundreds of proteins, often conserved across the eukaryotic lineage. Microtubule severing, one critical mechanism for reorganizing microtubules, is performed by ATP-hydrolyzing enzyme complexes, including KATANIN. One subunit of the KATANIN microtubule-severing complex is called p60. p60 has an N-terminal microtubule-interacting domain[1–3], and a highly-conserved *A*TPases *a*ssociated with various cellular *a*ctivities (AAA-type) domain for nucleotide binding and hydrolysis, and a C-terminal oligomerization subdomain required for self-interaction[4]. The p60 subunit alone has a relatively low binding affinity for microtubules but still forms a multimeric ring capable of ATP-dependent severing[1–3]. p60 is effectively recruited to microtubules by the p80 subunit both in vitro[5] and in vivo[6]. In plants, the KATANIN complex assembles and severs primarily at microtubule crossovers[7–10]. The p60 and p80 subunits together form heterodimers that assemble into a higher-order ring complex that grips the *β*-tubulin polyglutamate tail to generate the force for ATP-hydrolysis-dependent severing[11]. Katanin p60 interaction with the *β*-tubulin subunit has also been confirmed in vivo in *Caenorhabditis elegans*[12], where both p60 and p80 play essential roles in meiotic spindle formation[13].

In plants, *p60* mutants have been isolated from many forward mutant screens due to their striking vegetative phenotypes[14–18]. In *Arabidopsis thaliana*, *Cucumis sativa* (cucumber), *Marchantia polymorpha*, and *Oryza sativa* (rice), the p60 subunit is encoded by one gene[16,18–20]. In *A. thaliana*, the *p60* mutant has aberrant cell elongation, small organs, less cellulose, altered lignin accumulation, and disrupted cortical microtubule organization[21–26]. KATANIN also mediates rapid microtubule reorganization in response to environmental cues, affecting morphological changes in response to stimuli such as blue light and mechanical stimulation[27–29]. Similarly, the cucumber *p60* mutant has smaller fruits with fewer cells[18]. The *M. polymorpha p60*

[1]Department of Botany and Plant Sciences, Center for Plant Cell Biology, University of California, Riverside, CA, USA. [2]Dept of Agronomy, Purdue University, West Lafayette, IN, USA. [3]Department of Botany, School of Biology, Aristotle University of Thessaloniki, Thessaloniki, Greece. [4]Department of Medicine, Health and Medical University, Potsdam, Germany. [5]University of North Texas, Denton, TX, USA. [6]Present address: Bioinformatics and Biostatistics Core, Van Andel Institute, Grand Rapids, MI, USA. ✉e-mail: crasmu@ucr.edu

mutant is small with defects in microtubule organization, organ development, and fertility[20]. In rice, the *p60* mutant has small and round leaves, and shorter cells with aberrant cortical microtubule organization[16]. Despite the striking macroscopic defects, an understanding of how KATANIN contributes to growth has not been fully explored.

In addition to facilitating proper interphase microtubule organization, KATANIN-mediated severing contributes to remodeling microtubule arrays during mitosis and cell division. In non-plant organisms, KATANIN plays a role in spindle formation[30] and cytokinesis[31–33], while its role in plants is also relevant during premitotic stages. The preprophase band (PPB), a land-plant-specific microtubule array, accumulates prior to mitosis at the future division site[34,35]. *M. polymorpha p60* mutants frequently fail to form PPBs and generate multiple aberrant polar organizers[20]. *A. thaliana p60* mutants generate abnormal PPBs[20,24,36] including frayed PPBs, where one side of the PPB has less or poorly organized microtubules, as well as delayed PPB narrowing[24,36]. Although metaphase spindles are occasionally tilted[36,37], time-lapse imaging showed that misoriented spindles did not lead to altered final division plane positioning in petioles[24]. There are no abnormalities in the early formation of the cytokinetic structure called the phragmoplast, although it too is occasionally misoriented. Later, during phragmoplast expansion, *p60* mutants form a double-arrow-shaped phragmoplast, also observed in mutants lacking other microtubule-binding proteins[38,39], likely due to aberrant unseverable microtubule connections between the reforming nuclear envelope and phragmoplast microtubules[24,36]. Even though PPB narrowing, mitosis, and cytokinesis are slower in *A. thaliana katanin* mutants, no obvious division orientation defects are observed in epidermal petiole cells[24]. In contrast, ectopic or misoriented divisions in roots lead to altered cell fates affecting protoxylem, pericycle, and root hair cells[17,36,37,40]. Whether misoriented divisions in *katanin* mutants are due solely to misoriented PPBs or additional phragmoplast guidance defects is currently unclear[24,36,37,41]. We show here that maize symmetrically dividing *katanin* mutant cells have defective PPBs but do not have additional phragmoplast guidance defects.

Recent focus has further highlighted connections between the nucleus and the PPB, and consequently, the final location of the division[42,43]. The MAIZE LINC KASH AtSINE-LIKE2 (MLKS2) protein tethers the nucleus to the actin cytoskeleton. MLKS2 plays a critical role in chromosome segregation during meiosis, maintaining proper nuclear morphology and asymmetric division plane positioning[44]. Further, it contributes both to nuclear polarization via movement towards the correct location and the maintenance of nuclear positioning prior to subsidiary cell divisions in maize[45]. The nucleus polarizes before the PPB forms[46,47], often dependent on an intact actin[48] or microtubule cytoskeleton[49]. When the nucleus is polarized incorrectly or is not maintained near the future division site, the PPB forms in the incorrect location, leading to misoriented divisions[44,45]. Formation of incorrectly positioned PPBs and unpolarized nuclei occurs in subsidiary cell divisions in mutants lacking actin nucleators, suggesting that connections between actin and the nucleus are important for division positioning[50,51]. Similarly, when the nucleus is displaced by centrifugation after formation of a PPB, an additional new PPB forms near the nucleus[52]. Together, these data highlight the important role of the nucleus in positioning the PPB. We show here that the PPB also influences nuclear positioning.

Here, we characterize the function of two *Katanin p60* homologs in maize and define their contributions to plant and organ growth. We show that cell elongation defects, a G1 delay, and a reduction in cell numbers together account for smaller organs in the *p60* double mutant. Further, the *p60* mutants have defects in PPB formation. PPB positioning defects are correlated with defects in nuclear positioning such that the nucleus is typically aberrantly close to microtubule accumulation. Finally, rare symmetric division plane positioning defects are due to occasionally misoriented PPBs, not phragmoplast guidance defects.

## Results

### Identification of *katanin* p60 mutants in *Zea mays* L

The maize genome contains two *Katanin p60* genes which encode proteins that are ~97% identical at the amino acid level, Zm00001eb156560 (*Dcd3a* on Chromosome 3) and Zm00001eb360490 (*Dcd3b/clt1* on Chromosome 8): individually they are mostly redundant but loss of both alters proper growth, male and female fertility, and division positioning (Fig. 1, Supplementary Fig. 1). The *discordia3* (*dcd3*) mutant was identified by subsidiary cell division positioning defects in the leaf epidermis, similar to *dcd1*[53,54] and *opaque1/dcd2*[55–58]. While originally thought to be a single mutant, genetic analysis revealed that *dcd3* was a double mutant in both *Dcd3a* and *Dcd3b*. DNA sequencing identified a 1 base pair deletion in Zm00001eb360490 on Chromosome 8, *dcd3b-1*, which led to a frameshift-induced premature stop codon at amino acid 444 that entirely removes the oligomerization domain. This deletion is in a similar location to the 1 base-pair deletion of the *fra2* mutant, which generates a strong mutant phenotype in *A. thaliana*[17], suggesting that *dcd3b-1* is deleterious (Fig. 1a). Additionally, *dcd3* also contained another mutant allele mapped to Chromosome 3 at the Zm00001eb156560 locus, *dcd3a-1* that contains several missense mutations, including two nonconservative proline to serine substitutions within the microtubule binding domain of *Dcd3a* (Fig. 1a, substitutions: S62T, A81V, R93S, P113S, P127S, V171A, K188R, and D478E). The *dcd3a-1 dcd3b-1* mutant was backcrossed into B73 4 times. The deleterious *dcd3a-1* allele is identical to the allele from the maize inbred line CML228, and it was subsequently also independently identified in an enhancer screen described below.

From an independent EMS screen, a semi-dominant mutant, *Clumped tassel1* (*Clt1*), was identified by its short stature, poor fertility, and clumped tassel phenotype, meaning that tassels have dense spikelets (Supplementary Fig. 1). *Clt1* was backcrossed to B73 8 times. *Clt1* homozygotes are shorter than wild-type siblings, and do not generate ears or properly developed tassels (Supplementary Fig. 1). *Clt1* homozygotes contain cells that are more isotropic and often have division plane orientation defects, including misoriented subsidiary cell divisions (Fig. 1e, ~25% *n* ≥ 300 stomatal complexes from ≥3 plants). *Clt1/+* plants have an intermediate plant height (Fig. 1c and Supplementary Fig. 1) and generate poorly fertile ears, with ~ 20 kernels/ear (*n* = 26 *Clt1/+* ears, examples shown in Supplementary Fig. 1f). *Clt1* was previously mapped to an interval on the long arm of Chromosome 8[59–61]. Subsequent characterization identified a missense mutation (C to T at coding sequence position 1061 bp), changing amino acid 354 from serine to phenylalanine within the ATPase domain (Fig. 1a, Supplementary Fig. 2). This serine is conserved in >90% of 167 plant homologs examined, and the mutation has a Sorting Intolerant from Tolerant (SIFT) score of 0, where 0 is the most deleterious possible score[62]. While ATPase activity is predicted to be reduced or eliminated in p60[Clt1], ATP-binding, microtubule-, p80-, and self-interaction domains are intact. Our hypothesis is that p60[Clt1] generates non-functional complexes through interactions with the p60[Dcd3a], the p80 subunits, and other proteins. Heterozygotes (*Clt1/+*) contain 3 wildtype p60 isoforms: two p60[Dcd3a] and one p60[Dcd3b] while homozygotes (*Clt1/Clt1*) contain two wildtype p60[Dcd3a] and two mutant p60[Clt1]. Mutations within the AAA-type ATPase vacuolar protein sorting4 (*vps4*) at similar locations in the ATPase domain both prevent ATP hydrolysis and generate a dominant phenotype when expressed in wild-type yeast. The proposed mechanism for the dominant phenotype is that the mutant protein forms nonfunctional complexes (that cannot hydrolyze ATP) with the wild-type VPS4 protein and other subunits[63].

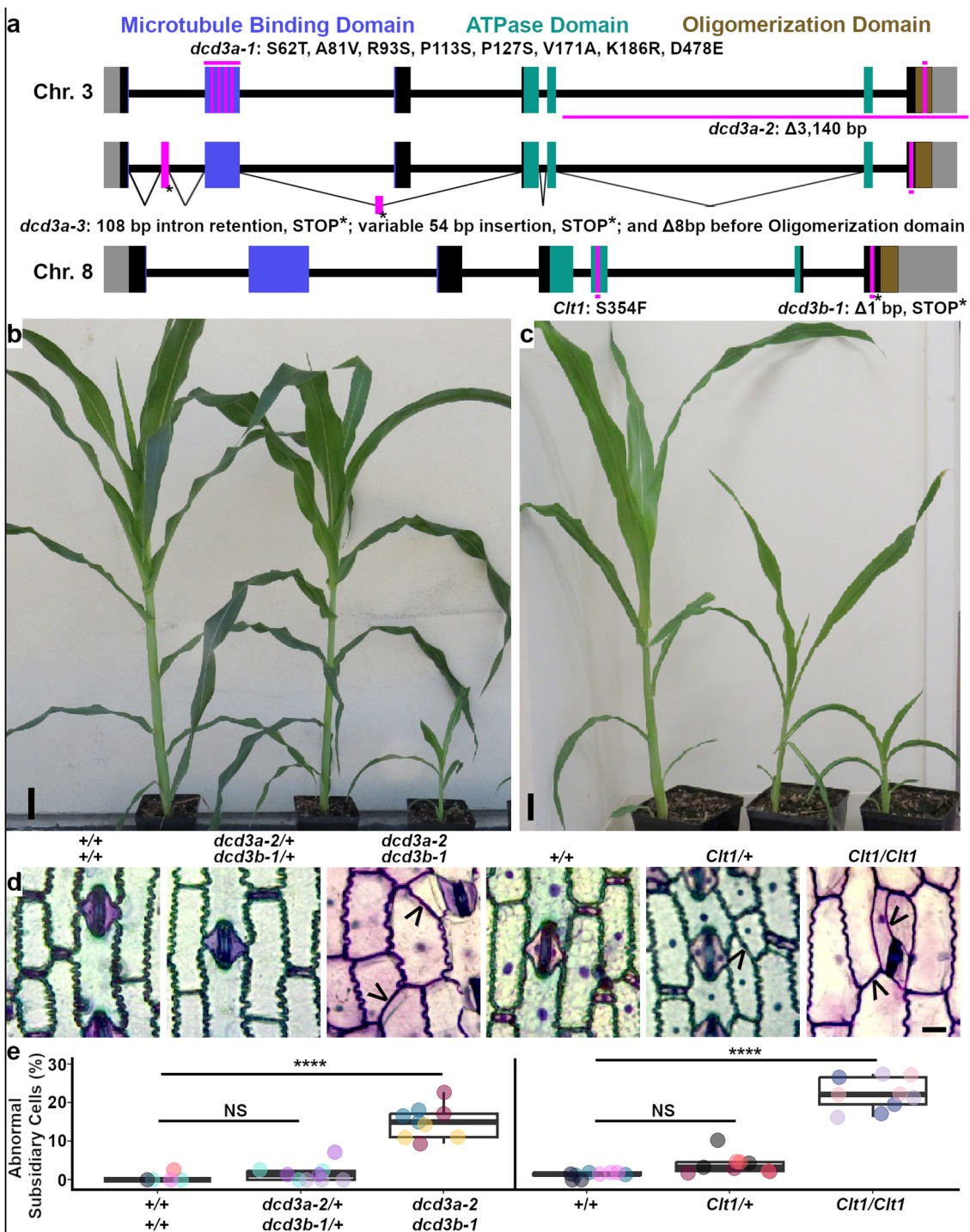

**Fig. 1 | Two *KATANIN p60* encoding genes in maize with two or three independent mutant alleles. a** Gene schematic depicting differences compared to B73. **b**, **c** Representative 4- or 5-week-old sibling plants. Scale bar = 5 cm. **d** Toluidine-blue-O-stained leaf 8 epidermal peels. Black arrowheads indicate potential division plane orientation defects. Scale bar = 10 μm. **e** Subsidiary cell division defect (%) from leaf 5 epidermal glue impressions from n ≥ 100 subsidiary cells each ≥ 3 plants per genotype. Different colors represent individual plants. Boxplots show median, quartiles, and whiskers ≤1.5X interquartile range. Two-sided Welch's pairwise t-test *p*-value **** <0.0001 with Bonferroni correction applied. Exact subsidiary cell numbers, plant numbers, and *p*-values are in Source Data.

To further clarify how *Clt1* functions, compound heterozygotes were generated that contained both *Clt1* and *dcd3b-1*, while wild type at the *Dcd3a* locus (+/+ *Clt1/dcd3b-1*). *Clt1/dcd3b-1* plants were small and infertile, similar to *Clt1* homozygotes and to *dcd3a-1 dcd3b-1* double mutants (Supplementary Fig. 3). The similarities in phenotypes support the hypothesis that p60^Clt1 assembles with other p60s to generate non-functional complexes. Fewer wild-type p60 subunits in the *Clt1/*

*dcd3b-1* compound heterozygote would lead to a higher proportion of non-functional complexes, finally leading to decreased fertility and growth.

The semi-dominant *Clt1/+* mutant was crossed to the Nested Association Mapping (NAM) parent lines to identify *Clt1/+* enhancers. Three enhancers significantly reduced *Clt1/+* plant height from maize inbred lines CML228, CML333, Ki3, and Ki11 (Supplementary

Fig. 4 and 5). These lines contained alleles of *Dcd3b* that were identical or very similar to B73 (see Source Data for sequences). The genic region containing the enhancers mapped to Chromosome 3, overlapping the location of *Dcd3a* (e.g., Supplementary Fig. 5). As described above, CML228 contains an alternative allele of *Dcd3a* (*dcd3a-1*), which has low *dcd3a* transcript accumulation compared to B73 and other inbred lines, similar to other alleles described below (Supplementary Fig. 4)[64,65]. Whether low transcript accumulation or amino acid substitutions in p60[Dcd3a-1] reduce microtubule severing (see below) is unknown. CML333 contains a deletion of 3,140 bp starting 8,690 bp from the start of the coding sequence in Zm00001eb156560; this mutant allele is termed *dcd3a-2* (Fig. 1a). The transcript analyzed by RT-PCR generated an early stop codon that removes most of the ATPase domain and all of the oligomerization domain, generating a likely null allele (Supplementary Fig. 4e, f). The *dcd3a* locus in Ki3 and Ki11 (*dcd3a-3*) contains an 8-bp deletion before the oligomerization domain and generates a variety of alternatively spliced products by RT-PCR (Fig. 1a, Supplementary Fig. 5f–h). Combining *dcd3a-3* with *Clt1/+* generated small plants lacking ears or viable pollen (Supplementary Fig. 5a, e). We were unable to generate homozygous *dcd3a-2 Clt1* double mutants (*n* > 10 crosses), but we were able to generate *dcd3a-2 dcd3b-1* double mutants. The *dcd3a-2 dcd3b-1* double mutants were backcrossed to B73 twice. Similar to *Clt1* homozygotes, *dcd3a-2 dcd3b-1* double mutants were sterile: double mutants did not generate ears, and tassels did not generate viable pollen (Supplementary Fig. 1i, n = 0/ 26 plants). Both homozygous *Clt1* mutants and *dcd3a-2 dcd3b-1* mutants also had reproducible differences in TBO staining (Fig. 1d, e), suggesting altered cell-wall composition, which may be similar to cell-wall alterations in Arabidopsis *katanin* mutants[21]. Additionally, the *dcd3a-2 dcd3b-1* mutants had short stature and division positioning defects (Fig. 1b, 1d). Symmetric division defects were rare (~5%) and are described in more detail below. The asymmetric divisions that generate subsidiary cells in stomatal complexes were also sometimes misoriented (~15%) in the *dcd3a-2 dcd3b-1* double mutant (Fig. 1e). Due to the complexity and sterility of possible mutant allele combinations, we primarily focused on the likely loss-of-function *dcd3a-2 dcd3b-1* double mutants.

### *p60* mutants have decreased microtubule severing frequency

The *dcd3a-2 dcd3b-1* double mutant has altered cortical microtubule orientation and anisotropy in the expansion zones of leaves and roots. Four-week-old and 8-day-old plants expressing YFP-TUBULIN were used to image microtubules in the leaf and root elongation zones, respectively. The direction of growth (Fig. 2a) was used to measure the anisotropy (directional uniformity) and orientation of microtubules. In leaves, *dcd3a-2 dcd3b-1* microtubule anisotropy was lower than wild-type siblings although not significantly different (Fig. 2b). In roots, *dcd3a-2 dcd3b-1* mutants had significantly decreased microtubule anisotropy (Fig. 2d), similar to Arabidopsis *katanin* mutants[7,21,24,66,67]. There was no difference in the highly variable microtubule angle with respect to the leaf growth axis (Fig. 2c; *n* ≥ 20 cells from ≥ 3 plants per genotype). In contrast, there was a significant decrease in microtubule angle in elongating root epidermal cells (Fig. 2e). *dcd3a-2* and *dcd3b-1* single mutants had similar microtubule anisotropy and angles to wild-type plants (Supplementary Fig. 6). Microtubule density of leaf epidermal cells was similar in all genotypes (Supplementary Fig. 6c).

For microtubule severing, we selected a subset of expanding leaf cells with similar cortical microtubule density and anisotropy to analyze (≥15 cells per genotype, 3 or more plants, Supplementary Fig. 7). Microtubule severing frequency was obtained by counting the number of microtubule severing events at microtubule crossovers per area over time (Fig. 2f). Wild type and double heterozygote *dcd3a-2/+ dcd3b-1/+* plants grew similarly and had similar severing frequencies (Fig. 2g), microtubule density and anisotropy (Fig. 2, Supplementary Fig. 6 a, c). This, together with sterility of the *dcd3a-2 dcd3b-1* mutant

plants, prompted us to use the double heterozygote as a wild-type control in subsequent experiments as the *dcd3a-2 dcd3b-1* double mutants were most often generated from *dcd3a-2/+ dcd3b-1 X dcd3a-2 dcd3b-1/+* or reciprocal crosses.

Both *dcd3a-2 dcd3b-1* double mutants and homozygous *Clt1* mutants had reduced microtubule severing compared to wild-type siblings. The *dcd3a-2 dcd3b-1* double mutants had >2-fold reduction in microtubule severing frequency when compared to wild-type siblings (Fig. 2g; Two-sided T-test *p*-value < 0.001). Additionally, *dcd3a-1 dcd3b-1* double mutants had a >1.3-fold reduction in microtubule severing frequency when compared to wild-type siblings (Fig. 2g, *p*-value = 0.0026). Surprisingly, there was no significant difference in microtubule severing frequency between *Clt1/+* and wild-type elongation-zone cells (Fig. 2g; *p*-value = 0.3). This was unexpected because the *Clt1/+* phenotype is macroscopically obvious, although less severe than double mutants or *Clt1* homozygotes (Supplementary Figs. 1 and 3). *Clt1* homozygous mutants had >2-fold reduction in microtubule severing frequency when compared to wild type, similar to the *dcd3a-2 dcd3b-1* double mutant (Fig. 2g, *p*-value < 0.001). While severing frequencies of several other mutant combinations tested were indistinguishable from wild-type, single *dcd3b-1* homozygous mutants were ~1.3-fold lower than wild-type siblings (Supplementary Fig. 8). Together, these data indicate that both p60 proteins significantly contribute to microtubule severing, but that p60[Dcd3b] plays a more prominent role, particularly in the elongation zone of adult leaves where severing was measured.

### The *dcd3a-2 dcd3b-1* double mutants have small roots and leaves

Similar to *katanin* mutants in other plants[15,16,21,68], *dcd3a-2 dcd3b-1* double mutants have reduced overall height (Fig. 1b, Supplementary Fig. 1) and ~3-fold smaller roots at day 4 with slower growth compared to wild-type siblings (Supplementary Fig. 9). We assessed V5 leaf elongation rates (n ≥ 15 plants each) and while wild-type leaf elongation rates were similar to previous reports[69–71] (2.8 mm/hr ± 0.2 mm SE), mutants elongated significantly slower (1.6 mm/hr ± 0.01 SE, Two-sided Student's *t*-test *p* value = 2.9e-6 see Source Data). Additionally, mutants also had smaller fully-grown leaves (Fig. 3). First leaves of 11-day-old mutant plants were ~3.5-fold smaller than corresponding wild-type sibling leaves (V1, Fig. 3c, Supplementary Table 1). Further, the *dcd3a-2 dcd3b-1* double mutants showed ~3-fold smaller leaves compared to wild-type leaves at other developmental stages: fully-expanded 5th leaves (V5, transitioning between juvenile and adult leaves) and fully-expanded adult leaves (V8) (Fig. 3f, i, Supplementary Table 1). The ~3-fold smaller leaves and roots, and slower growth rates, thus account for the smaller size of *dcd3a-2 dcd3b-1* mutants as compared to wild-type siblings.

### Small aberrantly expanded cells do not fully account for small leaves

To determine why mutants are smaller, we first tested the hypothesis that cell elongation defects caused by decreased microtubule anisotropy generate small leaves. Epidermal pavement cell and stomatal cell complex outlines were obtained from the base, middle, and tip of the leaf to measure cell areas (Supplementary Fig. 10, Supplementary Table 2). In leaves one (V1), five (V5), and eight (V8), pavement cells of the *dcd3a-2 dcd3b-1* double mutant were significantly less elongated (Fig. 3a, d, g) and smaller than wild-type siblings (Fig. 3b, e, h), similar to previous reports in other plants[16,21,26]. Using cell length and width parameters of the mutants (Fig. 3a, d, g), we generated a projection to reflect how this would alter wild-type leaf size (Fig. 3c, f, i: Projection 1, orange). Not surprisingly, these projections generated shorter and wider leaves, with a projected leaf area ~46-77% of wild-type leaves. However, the projections were still ~1.6–2.4-fold larger than the mutant leaves. This indicates that altered cell elongation partially contributes to the small leaves observed in *dcd3a-2 dcd3b-1* mutants. Next, we used

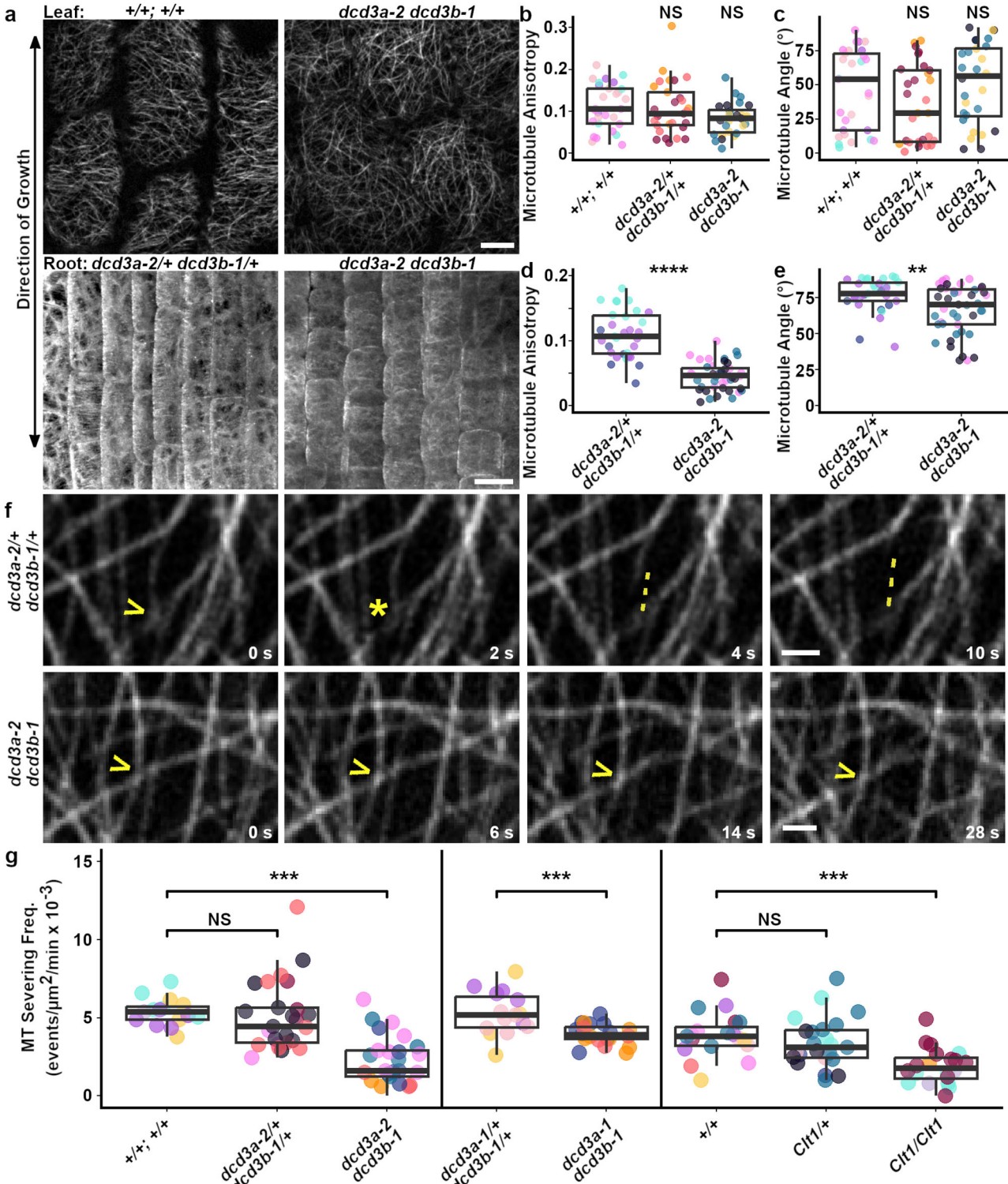

**Fig. 2 | *Katanin p60* mutants have reduced microtubule anisotropy and severing compared to wild-type siblings. Plants express YFP-TUBULIN.**
**a** Representative micrographs of interphase cortical microtubules aligned by growth axis in leaf 12 (+/+; +/+) or L6 (*dcd3a-2 dcd3b-1*) epidermal cells (top; scale bar = 10 μm) and root cells 10 days post sowing (bottom; scale bar = 20 μm). **b** Leaf microtubule anisotropy and **c** average microtubule angle with respect to the growth axis, *n* ≥ 22 cells from ≥ 3 plants per genotype. **d** Root microtubule anisotropy and **e** average microtubule angle with respect to the growth axis *n* ≥ 31 cells from ≥ 3

plants per genotype. **f** Representative micrographs showing microtubule severing. Yellow arrows = microtubule crossover. Yellow asterisk = severing event. Yellow dotted line follows the severing-induced depolymerization. Scale bars = 1 μm. **g** Boxplots of microtubule severing frequencies, each dot represents one cell, *n* ≥ 15 cells from ≥ 3 plants per genotype. All boxplots show median, quartiles and whiskers ≤1.5X the interquartile range. Two-sided Wilcoxon test (**b**–**e**) or two-sided Student's *t*-test (**g**), *p*-value NS > .05, * = 0.01, ** <0.01, *** <0.001. Exact cell numbers, plant numbers, and *p*-values are in Source Data.

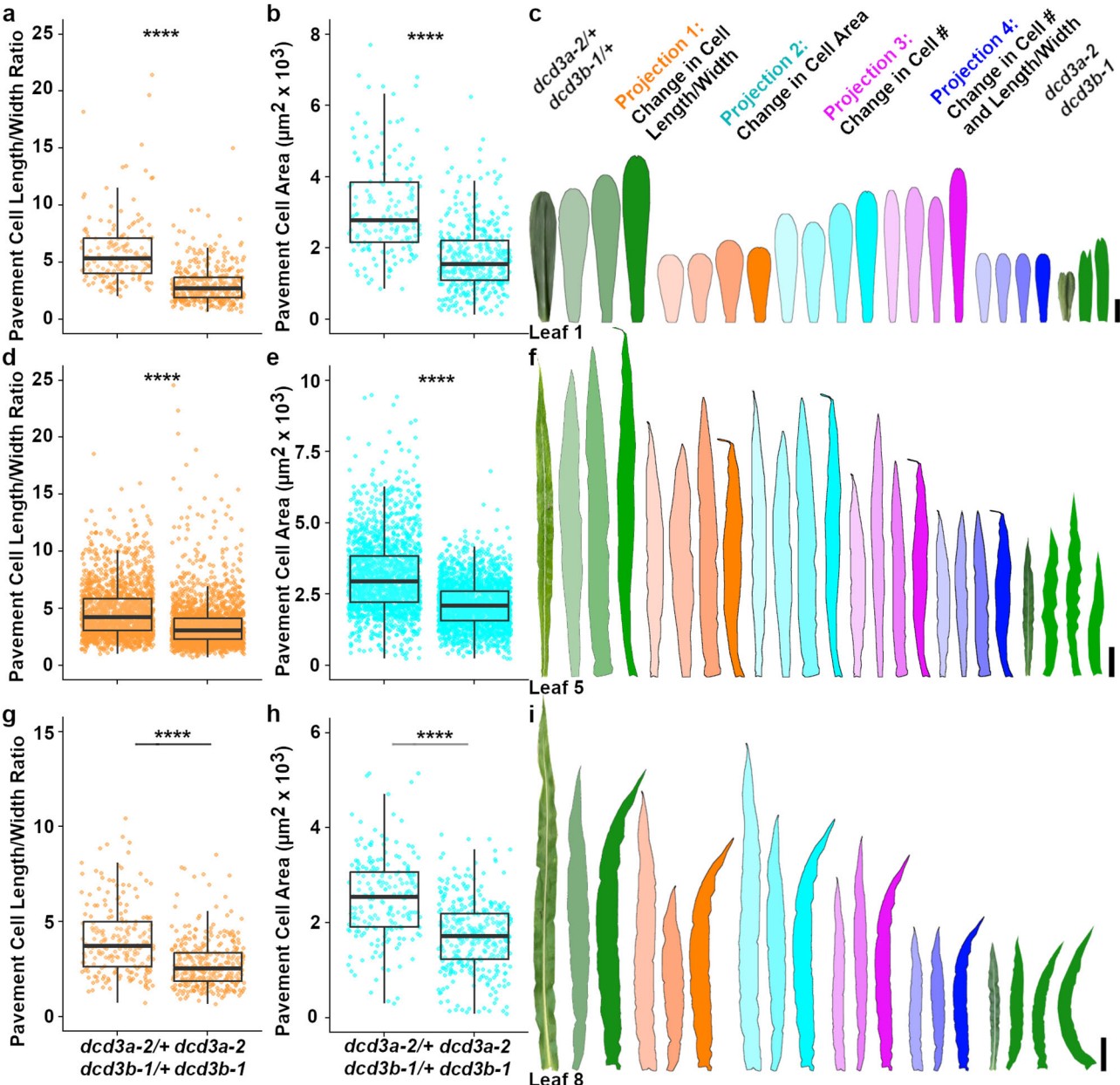

**Fig. 3 | *dcd3a-2 dcd3b-1* double mutants have smaller leaves due to fewer and smaller cells.** Pavement cell length/width ratio in wild-type *dcd3a-2/+ dcd3b-1/+* and mutant *dcd3a-2 dcd3b-1* for **a** leaf 1 **d** leaf 5 **g** leaf 8. Total cell area ($\mu m^2$ x $10^3$) of pavement cells for (**b**) leaf 1 (**e**) leaf 5 (**h**) leaf 8. $N \geq 3$ plants per genotype. Boxplots show median and quartiles with whiskers ≤1.5X the interquartile range. Leaf projections for (**c**) leaf 1, (**f**) leaf 5, and (**i**) leaf 8. Scale bars = 1, 5, and 10 cm, respectively. Two-sided Welch's pairwise t-test *p*-value < 0.00001 ****. Supporting data in Supplementary Tables 1, 2, Supplementary Fig. 10, and Source Data include plant number, cell number and exact *P*-values.

the average smaller cell areas of the *dcd3a-2 dcd3b-1* double mutants to generate Projection 2 (Fig. 3c, f, i, cyan). Projection 2 generated ~0.7–0.8-fold smaller projected leaves when compared to wild-type but ~2.2–2.7-fold larger than the mutants. Projection 1 and Projection 2 explicitly assume that wild-type and *dcd3a-2 dcd3b-1* mutants have the same number of cells within the leaf.

### *dcd3a-2 dcd3b-1* mutant leaves have fewer cells than wild-type leaves

Because smaller cell areas did not generate leaf projections that were similar to *dcd3a-2 dcd3b-1* mutant leaves, we next estimated cell numbers in wild-type and mutant leaves. We estimated the cell number based on extrapolations from the number of cells per area in several locations (see Materials and Methods). *dcd3a-2 dcd3b-1* mutants had

approximately 3-fold fewer cells in leaf 1 compared to wild-type siblings (Supplementary Table 1; T-test *p*-value = 0.002). Other mutant leaves (V5 and V8) also had ~2–2.7-fold fewer cells than wild-type siblings (Supplementary Table 1). We next used the average of these reduced mutant cell numbers to generate new projections of wild-type leaves (Fig. 3c, f, i: Projection 3, magenta). The projected leaves had ~42–58% the wild-type leaf area but were still between 143% and 197% larger than the mutant leaves (Fig. 3c, f, i). This suggests that reduced cell number contributes to the different leaf sizes. Finally, we generated a projection that combined both cell shape differences and reduced cell number, Projection 4 (Fig. 3c, f, i, blue). Not surprisingly, these projections generated leaves more similar to *dcd3a-2 dcd3b-1* leaves, with projected leaf areas ~101–114% the size of the average *dcd3a-2 dcd3b-1* leaves. Taken together, these results suggest that p60

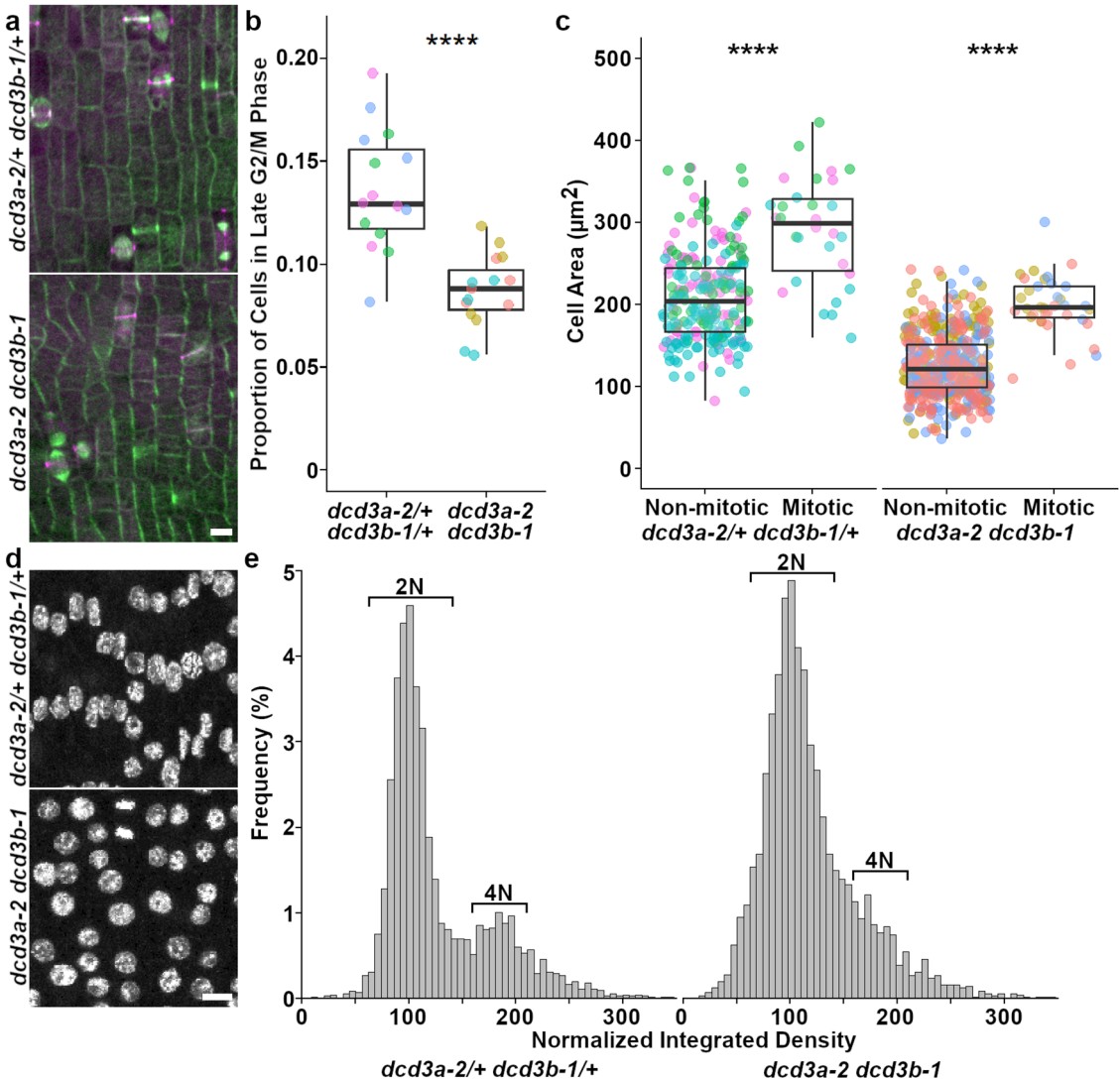

**Fig. 4 | The *dcd3a-2 dcd3b-1* double mutants have fewer actively dividing cells in the symmetric division zone and a delay in G1. a** Maximum-projection micrographs of the *dcd3a-2/+ dcd3b-1/+* and *dcd3a-2 dcd3b-1* symmetric division zone expressing CFP-TUBULIN (green) and TAN1-YFP (magenta). Scale bar = 10 μm. **b** Boxplot of the proportion of late G2/M phase, each dot represents one micrograph. Median and quartiles are shown and whiskers are ≤1.5X the interquartile range. Two-sided Welch's pairwise t-test *p*-value = 1.2e−5 (**c**) Cell area (μm²) in wild-type (non-mitotic *n* = 226 cells; mitotic *n* = 30 cells) and mutant (non-mitotic *n* = 430 cells; mitotic *n* = 36 cells). Dot colors represent individual plants, *N* = 3 plants per genotype. Two-sided Welch's pairwise t-test p-value = 1.8e−7 and 7.8e−15, respectively. Box plot shows median and quartiles with whiskers ≤1.5X the interquartile range. **d** Micrographs of stained DNA from -0.5 cm above the ligule in leaves. Scale bar = 10 μm. **e** Histograms showing the frequency of normalized integrated densities for 3 *dcd3a-2/+ dcd3b-1/+* and 3 *dcd3a-2 dcd3b* plants (n = 3076 and 4124 nuclei, respectively). G1, anaphase, and telophase are normalized to ~100 (2 N bracket) while G2, prophase, and metaphase are ~200 (4 N bracket).

plays a combined role in cell elongation and cell proliferation. Based on the relative reductions in leaf area, we estimated that cell elongation defects contributed slightly more than reduced cell proliferation to overall size.

### *dcd3a-2 dcd3b-1* mutants are delayed in G1 but not mitosis

To investigate whether fewer cells observed in *dcd3a-2 dcd3b-1* mutants were due to delays in cell division, we performed time-lapse analysis of symmetric cell divisions in four-week-old plants. Contrary to delays observed in Arabidopsis *katanin p60* mutants[24], transverse symmetric division duration measured from the start of metaphase (spindle formation) through phragmoplast disassembly was similar between *dcd3a-2 dcd3b-1* double mutants and wild-type siblings, as were metaphase and telophase durations (Supplementary Table 3). Similar to Arabidopsis *p60* mutants[24], phragmoplast morphology was altered in the *dcd3a-2 dcd3b-1* double mutants with more acute angles

and increased lengths (Supplementary Fig. 1i). Despite altered morphology, phragmoplast expansion rates between mutants and wild-type plants were similar (*dcd3a-2/+ dcd3b-1/+*: 0.31 μm/min ± 0.02 SE, n = 69 cells; *dcd3a-2 dcd3b-1*: 0.30 μm/min ± 0.02 SE n = 48 cells; Wilcoxon rank sum test *p*-value = 0.27). Similar cell division times between the double mutant and wild-type sibling suggest that the reduced cell number is not due to significant delays in metaphase through the end of cytokinesis.

As an independent test, we compared the relative proportions of specific division stages in symmetrically dividing zones (ligule height ~2 mm) using developmentally matched micrographs of *dcd3a-2 dcd3b-1* double mutant and wild type plants, assessed with a live cell marker for microtubules in late G2 and actively dividing cells (e.g., late G2/prophase, metaphase, or telophase, Fig. 4a). The cell-cycle stages represented in mutants and wild types were proportional to each other (Supplementary Table 4), indicating that *dcd3a-2 dcd3b-1* double

mutants do not have delays in any mitotic stage compared with wild-type siblings. That no particular mitotic stage is overrepresented is consistent with time-lapse imaging showing no significant difference in mitotic progression timing.

Although the proportions of specific mitotic stages were similar, significantly fewer cells in the symmetric division zone were actively dividing in mutants compared to wild-type plants. *dcd3a-2 dcd3b-1* mutants have ~1.6-fold fewer total cells in preprophase/prophase through cytokinesis when compared to wild-type siblings (Fig. 4b; Supplementary Table 5). Live-cell imaging and simulation experiments indicate that cell size and cell-cycle progression are linked[72,73]. To determine whether small cell size might be responsible for delayed entry into mitosis, we measured areas of mitotic and non-mitotic cells from mutant and wild-type dividing cells described above (3 plants each, from V8 and V12, respectively). *dcd3a-2 dcd3b-1* mutants had smaller cell areas in both mitotic and non-mitotic cells (Fig. 4c), with 1.6-fold larger mitotic cells. Wild-type mitotic cells were 1.4-fold larger than non-mitotic cells. As expected, cells entering division tended to be larger than non-mitotic cells. However, since the *dcd3a dcd3b-1* double mutant cells start small, it is likely that they also enter mitosis after significant delays in G1 or G2, or both, to overcome minimum cell-size thresholds.

To assess whether mutant cells were delayed in G1 or G2, we measured fluorescence intensities of DNA to reveal a significant delay in G1 in *dcd3a-2 dcd3b-1* double mutant. Nuclei from developmentally matched leaves of *dcd3a-2 dcd3b-1* and wild-type siblings were used to measure the integrated fluorescent densities (Fig. 4d, e; $n > 3000$ nuclei from 3 of each genotype). Wild-type siblings had clear peaks corresponding to 2 N (G1, and more rarely anaphase or telophase) ~69%, $n = 2108$ nuclei from bins 60–140) and 4 N (G2, prophase, and metaphase, ~18%, $n = 543$ nuclei from bins 160-210) In contrast, the *dcd3a-2 dcd3b-1* mutant nuclear integrated fluorescence densities were more frequently found in the 2 N peak (~73%, $n = 2,993$ nuclei, Chi-square $p$-value = 5.7e−4, see Source Data for exact values) and a smaller 4 N peak (~12%, $n = 492$ nuclei). Wild-type siblings also had signficantly less S-phase cells (~5%, $n = 153$ nuclei) than *dcd3a-2 dcd3b-1* mutants (~8%, $n = 313$ nuclei). Delayed G1 progression in the *dcd3a-2 dcd3b-1* double mutant is consistent with fewer overall cells.

## Misoriented PPBs lead to misoriented divisions

Similar to *katanin* mutants in other plants[24,36,40], the *dcd3a-2 dcd3b-1* double mutant has a high frequency of abnormal PPBs (~54.6%, $n = 98/182$ cells from 3 plants, Table 1). In contrast, most PPBs in wild-type siblings were normal (~91%, $n = 147/161$). PPBs classified as "normal" completely encircled the nucleus with consistent CFP-TUBULIN fluorescence intensity across the whole band (Fig. 5a). Uneven PPBs, where

the microtubule accumulation is not consistent across the whole band (Fig. 5a), occurred in ~32% of mutants but only ~5% in wild-type cells. While rarely observed, four-sided, three-sided, and split PPBs occurred with similar frequency in wild-type and mutant cells. The *dcd3a-2 dcd3b-1* mutant also had ~9% one-sided or incomplete PPBs, similar to Arabidopsis *katanin* mutants[24,36], and 5% misoriented PPBs, where the PPB was positioned perpendicular rather than parallel across in the cell (Fig. 5a) to generate a curved division. *dcd3a-2 dcd3b-1* mutants had ~46% normal PPBs (Fig. 5a). Neither one-sided nor misoriented PPBs were observed in wild-type siblings.

To determine if one-sided or uneven PPBs alter the timing of *dcd3a-2 dcd3b-1* symmetric divisions, we re-analyzed the time-lapse imaging from Supplementary Table 3. Rare one-sided PPB cells tended to have both more variable and delayed division times (Supplementary Table 6). Those cells with uneven PPBs also had more variable timing, but overall division times were not significantly different than cells with normal PPBs.

Next, one-sided or uneven PPB division trajectories were analyzed to show that they do not alter division plane positioning in *dcd3a-2 dcd3b-1* mutants (Fig. 5d–g, $n > 3$ plants). PPBs and division planes were positioned correctly in wild-type divisions ($n = 69/69$ cells, Supplementary Movie 1). Similarly, *dcd3a-2 dcd3b-1* double mutant cells with normal ($n = 17/17$ cells) and uneven ($n = 23/23$ cells, Supplementary Movie 2) PPBs were divided in a normal position. One-sided PPBs also generated apparently properly oriented symmetric divisions that bisect the cell in half ($n = 7/7$ cells, Fig. 5d-f, Supplementary Movie 3). Together, this indicates that in symmetrically dividing cells, final division plane positioning is not aberrant even when the PPB is only partially formed.

Misoriented PPBs in the *dcd3a-2 dcd3b-1* mutants generated similarly positioned misoriented divisions ($n = 11$ from 4 plants, Fig. 5g, Supplementary Movie 4). We were able to detect these rare divisions by using cells in metaphase and telophase with TANGLED1-YFP at the cell cortex as a proxy for the former PPB location, as previously described[54]. In total, we analyzed 48 symmetric divisions, starting with the PPB until the completion of cytokinesis: no divisions were observed where the phragmoplast went to a location that was different from the location of the PPB ($n = 0/48$). This time-lapse experiment indicates that the phragmoplast does not have a guidance defect and does not contribute to misoriented divisions in symmetrically dividing leaf cells. Finally, these results indicate that division plane defects in symmetrically dividing cells in the *dcd3a-2 dcd3b-1* double mutant are due to misoriented PPB positioning.

## PPB abnormalities influence nuclear positioning

While analyzing images, we observed that nuclei in *dcd3a-2 dcd3b-1* mutants were often aberrantly close to the PPB, particularly when the PPB was partially formed. In interphase, there was no significant difference in the nuclear positioning between mutant and wild-type cells (distance between the nucleus and cell centroids, Supplementary Fig. 12; $n > 54$ cells from 3 plants of each genotype; Wilcoxon test $p$-value = 0.7). However, when cells were in late G2/prophase, nuclear centroids in *dcd3a-2 dcd3b-1* cells were often further away from the cell center compared to wild-type nuclear centroids (Supplementary Fig. 12; $n > 112$ cells from 3 plants; Wilcoxon test $p$-value = 4.2E−07). This indicated that nuclear positioning defects were specific to cells with PPBs.

To determine whether the PPB influenced nuclear positioning, we obtained the normalized distance between the nucleus centroid and the PPB from micrographs of *dcd3a-2 dcd3b-1* and wild-type siblings (Fig. 5a–c). Similar to wild-type sibling nuclei, the nuclei of *dcd3a-2 dcd3b-1* mutants with normal and 3-sided PPBs were close to the cell center (Fig. 5a–c; $p$-value > 0.05). However, in cells with uneven, one-sided, or mispositioned PPBs, the nucleus was typically located closer to the more abundant microtubule accumulations in *dcd3a-2 dcd3b-1*

**Table 1 | Unbiased tiled imaging of PPBs in the symmetric division zone revealed increased frequencies of abnormal PPBs in the *dcd3a-2 dcd3b-1* mutants compared to wild-type siblings**

| PPB Type | Wild type (*dcd3a-2/+ dcd3b-1/+*) | Mutant (*dcd3a-2 dcd3b-1*) | *P*-value |
|---|---|---|---|
| Normal PPBs | 147 (~91%) | 84 (~46%) | 3.5e−20*** |
| 4-Sided PPBs | 1 (0.6%) | 1 (0.6%) | 1 NS |
| 3-Sided PPBs | 2 (1.2%) | 9 (~5%) | 0.7 NS |
| Split PPBs | 3 (~2%) | 5 (~3%) | 0.73 NS |
| Uneven PPBs | 8 (~5%) | 58 (~32%) | 5.7e−11*** |
| One-sided PPBs | 0 | 16 (~9%) | 3.2 e−5*** |
| Misoriented PPBs | 0 | 9 (~5%) | 4.0 e−3* |
| Total n | 161 | 182 | |

Three 4-week-old plants per genotype were used. Two-sided Fisher's exact test (with Bonferroni correction of 0.05/7 = 0.007) < 0.001 ***, * = 0.004, not significant = NS > 0.007.

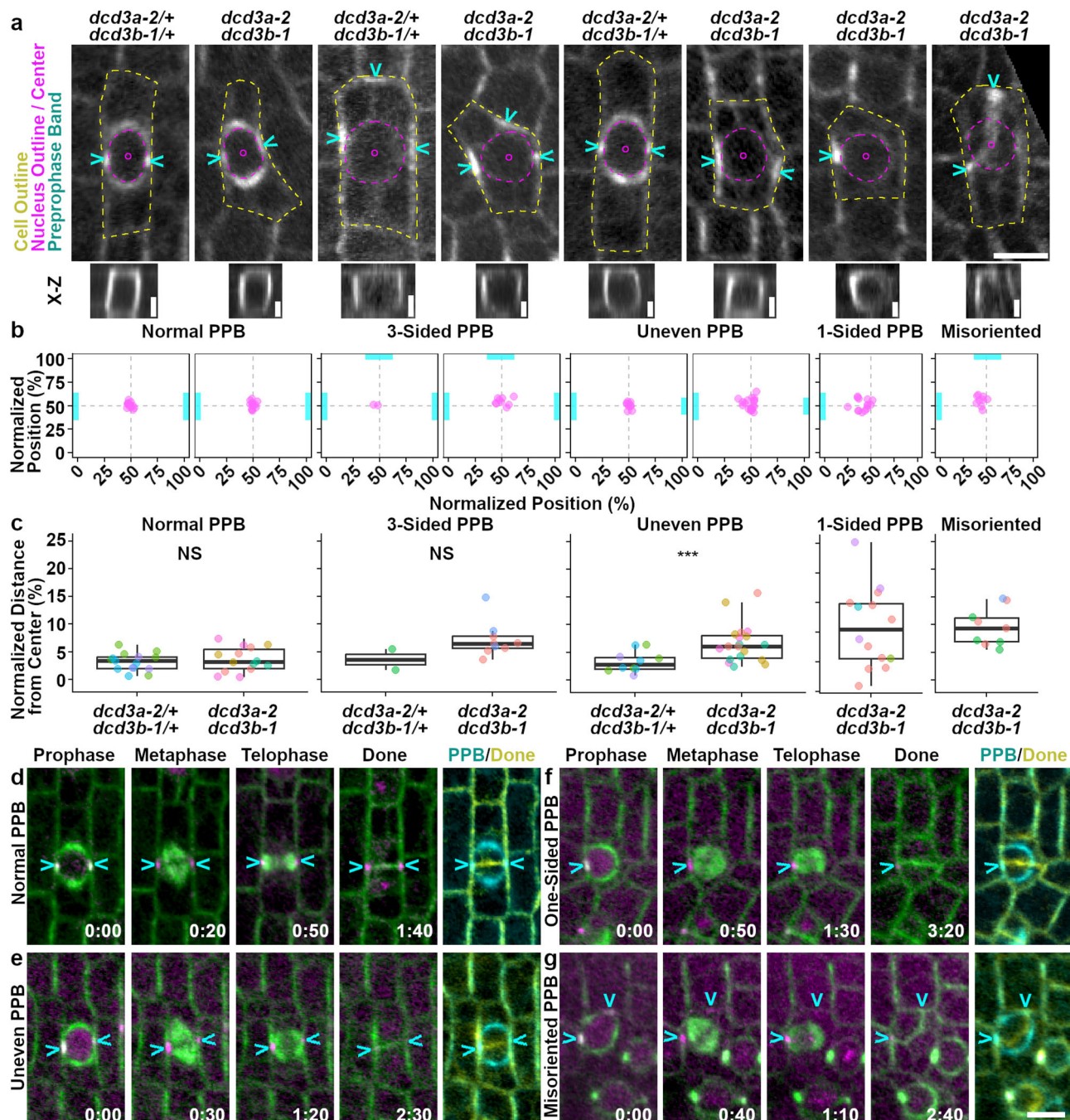

**Fig. 5 | Nuclear positioning defects in *dcd3a-2 dcd3b-1* mutants may be caused by aberrant PPB accumulation, while misoriented PPBs lead to misoriented divisions. a** Representative micrographs comparing nuclear positioning and PPBs in wild-type and mutant cells. Maximum Z-projection is used to highlight the misoriented PPB (far right). Cell outline = yellow dotted line. Nucleus outline and centroid = magenta. Cyan arrows = PPB. Representative X-Z maximum projections (scale bar = 5 μm) for each PPB type are shown below. For **b** and **c**, 3 or more plants per genotype were analyzed unless mentioned. **b** Normalized nuclear position (%). Each dot = one cell. Cyan bars indicate the PPB type and position. **c** Normalized distance of the nuclear center to the cell center (%). Each color = a plant. For normal PPBs: $n \geq 14$

cells per genotype; Two-sided Welch's pairwise t-test *p*-value = 0.52. Two plants per genotype had 3-sided PPBs; Two-sided Welch's pairwise t-test *p*-value = 0.09. For uneven PPBs, >10 cells; Wilcoxon rank sum test *p*-value = 4.08E-03. For 1-sided PPB: $n = 14$ cells. For misoriented PPBs: $n = 9$ cells. Exact cell numbers, plant numbers, and *p*-values are in Source Data. Boxplot shows median and quartiles with whiskers ≤1.5X the interquartile range. **d** *dcd3a-2/+ dcd3b-1/+* and **e–g** *dcd3a-2 dcd3b-1*, representative images CFP-TUBULIN (green) and TAN1-YFP (magenta) time-lapse. Time-lapse was obtained from n = 69 cells from 6 *dcd3a-2/+ dcd3b-1/+* plants and n = 48 cells from 8 *dcd3a-2 dcd3b-1* plants. Time indicated in hr:min. Scale bar = 10 μm.

cells (Fig. 5a, b), with a 2-fold or higher distance between nuclear and cell centroids (Fig. 5c, Wilcoxon rank sum, *p*-value = 4E−03). Together, these data suggest that PPB position influences the positioning of the nucleus in *dcd3a-2 dcd3b-1* mutants. While the PPB likely influences nuclear positioning, it is likely that the nucleus also influences PPB positioning. Further, despite no apparent nuclear positioning defects

during interphase in *dcd3a-2 dcd3b-1* mutants, we note that the nuclei were aberrantly round in the *dcd3a-2 dcd3b-1* mutant, with mean circularity of $0.95 \pm 0.001$ (mean ± SE, $n = 1718$ nuclei) versus the wild-type sibling mean circularity $0.92 \pm 0.002$ (mean ± SE, n = 1272 nuclei from *dcd3a-2/+ dcd3b-1/+*, Wilcoxon rank sum test *p*-value = 2.1e−93 from 3 plants each genotype, see Source Data).

## Discussion

KATANIN is conserved in many eukaryotes, except fungi. It is not found in bacteria or archaea. Similar to *A. thaliana*[7,9], maize KATANIN likely performs microtubule severing at crossovers due to reduced severing at crossovers in *dcd3a-2 dcd3b-1* mutants and *Clt1* mutants. In the future, it will be critical to test whether p60[Dcd3a], p60[Dcd3b], p60[Clt1], and p80 subunits directly interact to assess whether p60[Clt1] functions as an inactivating component of the multimeric p60-p80 complex. In other eukaryotic systems, Katanin-dependent microtubule severing does not occur at crossovers but instead at individual microtubules or bundles at the cell cortex in cultured *Drosophila melanogaster* cells[74], or to remove microtubules from centrosomes[5]. p60 copy number varies, from 1 in several plants e.g[16,18,20,21]., *Chlamydamonas reinhardtii*[75], and *C. elegans*[13] to 2 or more in mammals[30], *D. melanogaster*[74], and maize, as we showed here. Maize katanin p60 protein typically shares >90% identity with other monocot homologs, while it shares ~80% identity with *A. thaliana* and ~50% identity with the Human p60. Katanin often plays a fundamental role in fertility e.g[30,31,66,76–78]. which we also observed in *p60* maize mutants. Specific mechanisms contributing to fertility are distinct within organisms and highlight the varied role that p60 plays in microtubule reorganization.

Multiple independent analyses across several developmentally distinct leaves and roots with different growth conditions showed that *dcd3a-2 dcd3b-1* double mutants generated small, aberrantly-shaped organs. Further, leaf and root elongation rates were reduced. Small, less elongated cells are a hallmark of *p60* mutants in other plants[15,16,18,21] but how cell shape defects contribute to overall growth had not been analyzed in detail. Here, we have shown that both cell elongation and reduced frequency of cell division contribute to small organs, although cell elongation likely plays a dominant role.

Cell size thresholds contribute to the switch between G1 and S phases of the cell cycle[73] and also between G2 and M[72]. Here, we show that the *dcd3a-2 dcd3b-1* mutant has a delay in G1, with a corresponding reduction in G2/early M cells. We speculate that the G1 delay observed in *dcd3a-2 dcd3b-1* double mutant cells may be the primary driver for reductions in the number of actively dividing cells, since division itself was not delayed. Observations from Arabidopsis shoot apical meristems show that additional time is typically required for small cells to complete the cell cycle[72], despite smaller cells growing more quickly than larger neighboring cells[79]. Interestingly, blocking Arabidopsis meristem cells at or before S phase with the DNA synthesis inhibitor aphidicolin also prevents cell expansion, highlighting the interconnections between cell cycle progression and cell elongation[80]. In the future, it would be interesting to use recently developed live-cell markers to directly measure G1-S-G2-M transitions and proportions in live cells[72,81].

In our study, *katanin p60* mutant combinations typically had ~2-fold reduced microtubule severing frequencies compared to wild-type siblings. However, microtubule severing in maize epidermal cells was not completely eliminated, in contrast to microtubule severing frequencies of the *p60* mutant in the Arabidopsis petiole[24], and cotyledons[7,82]. Similarly, *p80* mutants reduced severing frequency ~11-fold in the Arabidopsis stamen[83]. One hypothesis for low but detectable levels of severing in mutants is that some allele combinations may reflect only partial loss of function. For example, *dcd3a-1*, the allele from the inbred CML228, contains many SNPs that generate several deleterious amino acid substitutions but no stop codon. Further, *dcd3a-1* generates a transcript, suggesting that it also generates a protein (Supplementary Fig. 4d). Relative reductions in severing are smaller in *dcd3a-1 dcd3b-1* mutants compared to *dcd3a-2 dcd3b-1*, suggesting that this mutant combination may reflect a partial reduction in activity.

An alternate and more likely hypothesis for residual severing in other mutant combinations is that other microtubule severing proteins may contribute more to severing in some monocots, such as maize and rice. For example, spastin plays a significant role in rice plant growth[84], while its role in maize is unknown. In contrast, Fidgetin-like in maize has no described vegetative role[85]. Determining whether severing still occurs in maize *katanin* mutants due to contributions from other microtubule severing proteins or due to residual partial function in maize awaits further study.

A significant fraction of PPBs in the *dcd3a dcd3b-1* double mutant were aberrantly formed, generating uneven PPBs and occasional frayed or, in rare cases, one-sided PPBs, similar to those observed in Arabidopsis roots and petioles[24,36,40]. Time-lapse analysis showed that failure to form a morphologically normal PPB led to division delays, similar to delays in division seen in Arabidopsis *katanin* mutants[24]. The rarity of one-sided PPBs likely contributes to overall similarity in division times between mutant and wild-type cells.

The majority of PPBs, even those that are aberrantly formed, appear correctly localized and generate properly oriented divisions in symmetrically dividing cells of the *dcd3a-2 dcd3b-1* double mutant. That poorly formed or undetectable PPBs do not cause major division positioning defects has also been observed in the Arabidopsis *tonneau recruiting motif* (*trm678*) triple mutant, as well as in the *tonneau1a* single mutant, which has greatly compromised PPBs but variable division positioning defects depending on cell type[86,87]. In contrast, lack of PPB formation is tightly correlated with mispositioned division in asymmetric, subsidiary cell divisions[54]. Consistently, asymmetric subsidiary cell divisions were mispositioned ~15% in *dcd3a-2 dcd3b-1* mutants, while symmetric division positioning defects occurred 3X less frequently (~5%). Together, this suggests that the PPB may play differential roles in division positioning in different cell types, with its role in division positioning more clearly revealed in asymmetric divisions.

Rare misoriented PPBs in symmetrically dividing cells of the *dcd3a-2 dcd3b-1* double mutant generated aberrant small cells with curved cell walls. In Arabidopsis roots, ectopic and mispositioned longitudinal divisions within *katanin* mutants generated additional protoxylem and calyptrogen cells that form additional xylem and root cap cells respectively[40]. Additionally, *katanin* mutants also generated mispositioned hair and non-hair cells in roots. Interestingly, aberrant cell-type specification was seen in cell files with apparently normal division planes, suggesting that, although the mutant has misoriented division plane positioning, that alone did not cause the misspecified cell fates[17]. Similar root hair and division mispositioning occurs in the *sabre* mutant[88,89]. Another fascinating division positioning defect generates a similar small and aberrantly shaped cell in ectopically dividing endodermal cells in the *inflorescence and root apices receptor kinase (irk)* mutant[90]. These small cells eventually transition into cortex cells. Whether the PPB is mispositioned in the *irk* mutant is not yet known.

Our examination of nuclear positioning defects highlights an important role of both the PPB and KATANIN in correctly orienting the nucleus during late G2/prophase. Connections between the PPB and the nuclear envelope have been observed in many organisms: these "bridge microtubules" are thought to promote bipolar spindle formation[20,91–94]. Interestingly, drugs that inhibit microtubule depolymerization disrupt mid-cell nuclear positioning in cells with PPBs, indicating that dynamic microtubule organization is critical for nuclear positioning[95]. Here, we showed that defects in PPB morphology were correlated with mispositioned nuclei: nuclei were located closer to more microtubule accumulation in *katanin* mutants. KATANIN-GFP accumulates at the nuclear envelope and the PPB during prophase/G2 with a proposed role in severing microtubules to establish bipolarity of the prophase spindle[36]. Here we suggest that it may also sever microtubule connections at the nuclear envelope to adjust nuclear position in relation to the PPB. The feedback between PPB formation and stabilization and nuclear positioning is likely to be complex. In addition, increased circularity of interphase *dcd3a-2 dcd3b-1* mutant nuclei

suggests that the connection between the cytoskeleton and the nucleus is disrupted in the *katanin* mutant. Similar aberrantly round nuclei have also been observed in mutants that lack connections between the cytoskeleton and the nucleus[96,97]. Some examples include *kaku1* (Myosin XI-i)[98], the maize *mlsk2*[44] and *crowded nuclei (crwn)* mutants[99,100]. The connections between nuclear shape and positioning in *p60* mutants are an interesting potential future direction.

# Methods

## Plant growth conditions

To generate plants with specific genotypes, maize plants were grown to maturity and crossed in standard field conditions at UCR, the University of North Texas, and Purdue University.

Maize plants were grown in the greenhouse with the following conditions: 1 L pots with soil (20% peat, 50% bark, 10% perlite, and 20% medium vermiculite) supplemented with magnesium nitrate (50 ppm N and 45 ppm Mg), calcium nitrate (75 ppm N and 90 ppm Ca), and Osmocote Classic 3-4 M (NPK 14-14-14 %, AICL SKU#E90550). Plants were grown under standard greenhouse conditions (~31–33 °C with supplemental lighting from 5 to 9 PM at ~400 μEm$^{-2}$s$^{-1}$). For analysis of the fully expanded fifth and eighth leaves, plants were grown for 5 weeks and 12 weeks, respectively. For time-lapse and other imaging, plants were grown for 4 weeks.

For first leaf and root growth analysis, germination paper was used to germinate kernels after sterilization according to[101,102] and as described below, and grown for 10-11 days in a growth chamber (Percival) at 24 °C with 16-h white light ~111 μE*m$^{-2}$s$^{-1}$ (F17T8/TL741 Fluorescent Tube (Philips)) and 8-h dark cycles.

For leaf elongation rate experiments, maize kernels were germinated in 1 L pots as described above and grown in a growth chamber (Percival PGC-40L2) at a constant temperature of 32 °C with 16-h light (~400 μEm$^{-2}$s$^{-1}$) and 8-h dark cycles.

## Mapping *Clt1* to a lesion in an ortholog of AtKTN1 on Chr 8

*Clt1* was mapped to an interval on the long arm of chromosome 8[59] that proved later to be ~21.8 Mb in length. To refine the interval for *Clt1* further, the mutant, which was generated by Dr. Gerry Neuffer using ethyl methanesulfonate (EMS) in an unknown genetic background, was introgressed for eight generations into the B73 inbred maize background. We defined a 6.6 Mb, non-B73 linkage block on Chr. 8, flanked on both sides by homozygous B73 loci, containing *Clt1* (Supplementary Fig. 2). Genotyping 259 more individuals from a B73 X *Clt1*/+ (B73) cross identified a recombinant that reduced the *Clt1* interval further with a new right flanking marker, ss230245925, at position 150,879,259 bp (Supplementary Fig. 2). No polymorphic markers could be identified to demarcate the left boundary of the non-B73 linkage block; the closest markers (ss230245196 and ss230245225) were homozygous for the B73 allele. We conservatively estimated marker ss230245130 as the putative left boundary instead, leaving an interval containing 16 genes that might underlie *Clt1*.

To identify the causative lesion within these 16 genes, genomic DNA from two *Clt1* homozygotes in a B73 background was sequenced. After aligning the reads to the B73 genome sequence (AGPv3.21; 6.4X and 6.7X coverage, respectively) and filtering out low-quality variant calls, plotting the number of non-B73 sites in genomic windows along Chr 8 (Supplementary Fig. 2) confirmed the interval obtained through positional cloning. The only mutation within the interval that affected the coding sequence of any gene was a single missense mutation in an *AtKTN1* ortholog, ZEAMMB73_183479 (also known as GRMZM2G017305 at Chr 8:150,792,636-150,799,106 AGPv3; Supplementary Fig. 2). The location of this gene coincided with a region of sharply reduced frequency of variant sites. The putative mutation had a Phred-scaled P-value from Fisher's exact test for strand bias (FS) of 0, indicating no strand bias during sequencing. The mutation is a C to T transition, which would change a serine to phenylalanine in the ATPase domain of the protein (Fig. 1a), predicted to be deleterious (SIFT score of 0). This SNP was confirmed by both a dCAPS marker and Sanger sequencing, which also showed that there is 100% identity with B73 at all other positions in this gene, including the 5′ untranslated region (UTR), exons, introns, and the portion of the 3′ UTR that was sequenced. These results strongly suggest that mutation of ZEAMMB73_183479 is the cause of *Clt1*.

## Genotyping

DNA was extracted from leaves or kernels using standard protocols[103,104]. Mutant alleles (*dcd3a-1*, *dcd3a-2*, *dcd3b-1*, *Clt1*) and/or transgenes (YFP-α-TUBULIN, CFP-β-TUBULIN, TAN1-YFP) were genotyped using 0.5 μM primers in Supplementary Table 7. Restriction enzyme digests followed PCR with SspI, MspI, and DraI for *dcd3a-1*, *dcd3b-1*, and *Clt1*, respectively. Presence of transgenes was also verified through painting leaves with 4 g/L glufosinate (Finale, Bayer) in 0.1% Tween 20 (Sigma) and scoring for resistance after 5-7 days.

## Microtubule Severing

Four- to five-week-old plants expressing YFP-α-TUBULIN or CFP-β-TUBULIN were dissected to the youngest emerging leaf. Tissue samples were obtained from the elongation zone (7 cm from the ligule for wild-type siblings and 4 cm for *dcd3a-2 dcd3b-1*) and mounted on water in a rose chamber as described[105]. Samples were imaged every 2 seconds. Time-lapses were bleach corrected (Simple Ratio) and re-aligned using the Translation selection in StackReg in FIJI[106]. Only cells with similar microtubule density and anisotropy were analyzed. Individual cells were selected with the Rotated Rectangle Tool, and the cell area was measured. Microtubule density and anisotropy were calculated for the first time frame of bleach-corrected (Simple Ratio) images using BoneJ[107] and FibrilTool[108] plugins, respectively, in FIJI. For microtubule density, images were thresholded using default segmentation prior to obtaining the Area/Volume Fraction using the BoneJ plugin. Microtubule density was calculated as 1 - Area/Volume. For microtubule severing analysis, a microtubule severing event was identified by the depolymerization of one or more microtubules at a microtubule crossover.

## Root and leaf growth experiments

Maize kernels were sterilized according to refs. 109,110, with the following modifications. Maize seeds were incubated in 80% ethanol for 3 minutes, rinsed with sterile water, incubated with 50% Bleach for 15 minutes, and rinsed with sterile water. Kernels were imbibed overnight in sterile water with rocking. Growth assays were conducted following the published method[102]. Sterilized maize seeds germinated on germination paper (Fisher Scientific #NC1466201) soaked in 2.5 g/L of Captan 50 fungicide (Southern Ag #01600), rolled and stored vertically in beakers filled with 400 mL of 0.5X pH5.7 Linsmaier and Skoog media (Caisson Labs #LSP03-50LT). The media was changed every 2 days. Root images with a 30.5 cm ruler were taken every 24 hours for 10 days with a Canon PowerShot SX540 HS digital camera. Germination was determined by the protrusion of the radicle from the coleorhiza, and root measurements excluded the coleorhiza. Root lengths were measured manually using FIJI.

Maize kernel DNA was genotyped[104] and two wild-type and two mutant kernels were planted in the growth chamber every day for 10 days. Upon leaf 5 emergence, leaf 5 length was measured using a 30.5 cm ruler every 24 hours for 7 days to obtain the leaf elongation rate[111].

## Microtubule anisotropy and orientation in roots

For microtubule anisotropy and orientation analysis in roots, maize kernels expressing YFP-TUBULIN were grown on germination paper as described above. At 10 days post-sowing, maize seedlings were imaged with a Leica SP8 confocal microscope. Elongation zone cells (captured

just below root hairs) were imaged 0.5 cm from the root tip in *dcd3a-2/+ dcd3b-1/+* and (-0.2 cm) in *dcd3a-2 dcd3b-1*. Microtubule anisotropy was obtained using the FibrilTool plugin in FIJI, and microtubule orientation (angle) was obtained by measuring the angle between the average microtubule orientation and the growth axis of the root.

## Leaf area and cell measurements

Fully expanded leaves 5 and 8 were obtained from at least 3 *dcd3a-2/+ dcd3b-1/+* and *dcd3a-2 dcd3b-1* mutant plants grown in the greenhouse. Leaf 1 was obtained from seedlings grown in germination paper rolls for 11 days. Whole leaf images were taken next to a 30.5 cm ruler using a camera (Canon PowerShot SX540 HS). Glue impressions[112] were made from the base, middle, and tip regions of the leaves and imaged with a Nikon light compound microscope using a 4X or 10X objective with a 2X AmScope camera attachment. Leaf areas, pavement cells, and stomatal complexes were outlined manually in FIJI.

Approximating Total Cell Counts in Leaves 1, 5, and 8:

The number of cells in a defined area was used to approximate the number of cells in the whole leaf. The defined areas consisted of at least 3 micrographs each from the tip, middle, and base of the leaf. Averages for each section were calculated separately and then combined as follows. PC = total pavement cells measured; SC = total stomatal complexes measured; PCA = pavement cell area; SCA = stomatal complex area.

$$Total\ Cells \approx \left[\frac{Total\ Leaf\ Area}{3}\right]\left[\frac{PC_{Base}+4\left(SC_{Base}\right)}{\sum PCA_{Base}+\sum SCA_{Base}}\right]$$
$$+ \left[\frac{Total\ Leaf\ Area}{3}\right]\left[\frac{PC_{Middle}+4\left(SC_{Middle}\right)}{\sum PCA_{Middle}+\sum SCA_{Middle}}\right]$$
$$+ \left[\frac{Total\ Leaf\ Area}{3}\right]\left[\frac{PC_{Tip}+4\left(SC_{Tip}\right)}{\sum PCA_{Tip}+\sum SCA_{Tip}}\right]$$

**Leaf projections.** Projection 1 was calculated by multiplying the average cell length and width of *dcd3a-2 dcd3b-1* mutants by the estimated number of cells along the long and short axes of each *dcd3a-2/+ dcd3b-1/+* leaf. Projection 2 multiplied the average cell area of *dcd3a-2 dcd3b-1* mutants by the estimated cell number for *dcd3a-2/+ dcd3b-1/+* leaves. Projection 3 multiplied the average estimated cell number of *dcd3a-2 dcd3b-1* mutants by the cell dimensions of each *dcd3a-2/+ dcd3b-1/+* leaf. Projection 4 multiplied the average estimated cell number by the average cell area of *dcd3a-2 dcd3b-1* mutants. All projections were made to scale in GIMP.

**Cell division and nuclear positioning analysis.** 4-week-old plants were used to obtain a leaf with a ligule height <2 mm. Samples were taken within 0.5 cm of the ligule of the emerging leaf. The abaxial side was mounted in water in a rose chamber.

Time-lapse of symmetric cell divisions of plants expressing either YFP-TUBULIN, or CFP-TUBULIN, and TAN1-YFP were obtained in 10-minute intervals. For metaphase time, the duration of the first frame without a PPB to the last frame before anaphase was counted. For telophase time, the duration of the first frame with a phragmoplast to the frame up to phragmoplast disassembly was counted. For phragmoplast expansion time, the first frame with a phragmoplast to the frame before the phragmoplast reached the cortex of the cell was counted.

**Nuclear integrated density measurements.** Three 4-week-old *dcd3a-2/+ dcd3b/+* and three *dcd3a-2 dcd3b-1* plants were dissected down to the 12th and 7th leaves, respectively, and samples were taken -0.5 cm above the ligule. Samples were fixed in a 1:3 (v/v) acetic acid: ethanol solution and rehydrated in 75%, 50%, 25% ethanol, and then deionized water. Samples were stained with 2 µM Vybrant DyeCycle Green

(ThermoFisher #V35004) for 3 hours in the dark at room temperature. Samples were rinsed with water and then loaded onto a Leica SP8 confocal. Z stacks covering the entire nuclei were gathered. To analyze the fluorescent integrated density, maximum projections were made, and the ImageJ plugin StarDist[113] was used to segment nuclei using the following options: versatile nuclei with Probability/Score Threshold of 0.5 and Overlap Threshold of 0.24. After manual correction, normalization was performed per image by dividing each fluorescent integrated density measurement by the most frequent image-specific peak density (2 N peak) before plotting as a histogram in R. The number of nuclei/DNA in 2 N (anaphase, telophase, and G1) and 4 N (G2, prophase, and metaphase) peaks was obtained by matching the shape of the curve in the normalized bins from 2 N (60–140) and 4 N (160–210), respectively. The number of nuclei in S was estimated by adding the remaining nuclei between 2 N and 4 N.

## Confocal microscopy

Images were acquired using a Yokogawa W1 spinning disk microscope on a Nikon Eclipse TE inverted stand with an EM-CCD camera (Hamamatsu 9100c) built by Solamere Technology Inc. Solid-state Obis lasers (40-100 mW) were used in combination with standard emission filters (Chroma Technology). For YFP-TUBULIN and TAN1-YFP, a 514 nm laser with an emission filter of 540/30 nm was used. For CFP-TUBULIN, a 445 nm laser with an emission filter of 480/40 nm was used. A water immersion objective (60X/1.2 NA) and oil objective (100X/1.45 NA) were used. Images and time-lapses were taken with Micromanager-1.4 using a 3-axis DC servo motor controller and ASI Piezo Z stage. For microtubule severing time-lapse, 2 second time intervals were used. For cell division time-lapse, 10-minute time intervals were used. When Z-stacks were acquired, a Z-interval of 2 microns was used.

For microtubule severing, some images were collected using a Zeiss LSM 880 confocal laser scanning microscope (100X oil objective immersion lens, NA = 1.46) with Airyscan. A 514 nm laser with bandpass filters of 465-505 nm with a long-pass 525 filter was used with a time interval of 2 seconds. Images were processed using default Airyscan settings with Zen software (Zeiss).

For maize roots expressing YFP-TUBULIN and leaf samples stained with Vybrant DyeCycle Green, images were collected on a Leica SP8 upright confocal microscope with a water immersion objective (40X/1.1 NA). For roots expressing YFP-TUBULIN, 514 nm excitation with emission of 515–589 nm was used. For Vybrant DyeCycle Green-stained leaves, 488 nm excitation with emission of 493–560 nm was used. Z-stacks were acquired with a 1 micron interval.

## Figure preparation

Figures were made using Gnu Image Manipulation Program (GIMP, v2.10.38, https://www.gimp.org/) or some combination of PowerPoint, Inkscape, and GIMP for Supplementary Figs. 1–5. Levels were adjusted linearly. If applicable, images were enlarged or rotated with no interpolation.

## Graphs and Statistics

Graphs and statistics were generated using R and RStudio using the following packages: tidyr[114], ggplot2[115], ggprism[116], paletter[117], and stringr[118].

## Reporting summary

Further information on research design is available in the Nature Portfolio Reporting Summary linked to this article.

## Data availability

The sequencing reads for *Clt1-1* homozygotes have been deposited at NCBI SRA under accession code PRJNA344581 https://www.ncbi.nlm.nih.gov/bioproject/PRJNA344581. Source data are provided with this paper.

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

## Acknowledgements

Thanks to Dr. Gerry Neuffer (University of Missouri) and the Maize Genetics Cooperation for generating and maintaining the *Clt1* mutant, respectively. Thanks to Drs. Laurie Smith and Kim Gallagher for generating the *dcd3* mutant, and Dr. Nicholas Miles for propagation. Thanks to Dr. Michael McCarthy (Radical Research LLC) for expert R advice. Thanks to Dr. Jaimie Van Norman (UCR/UCLA) for the use of the Leica. This work was supported by NSF grants (CAREER-IOS-1350874 AJW,

CAREER-MCB-1942734 CGR, MCB-2426623 CGR, DBI-0822495 KL and CW, DBI-1922642 SEM, REU-2051131 CH, REU-2447384 ID), a training grant to Purdue University Agronomy from USDA-NIFA and the Purdue Research Foundation to KL and CW, undergraduate research funding from UCR-RISE to CI, CH, IC.

## Author contributions

Conceived and designed the analysis: A.J.W., C.G.R., C.W., S.E.M., K.L., E.P. Collected the data: S.E.M., K.L., L.A.A., C.I., C.H., I.D., I.C., P.B., A.J.W., C.W., C.G.R. Contributed data or analysis tools: Performed the analysis: S.E.M., K.L., C.G.R., A.J.W. Wrote the paper: C.G.R., S.E.M., K.L., C.W., A.J.W. Revised and reviewed the paper: All authors reviewed the paper Funding acquisition: S.E.M., C.I., C.H., I.D., I.C., A.J.W., C.W., C.G.R.

## Competing interests

The authors declare no competing interests.
