## [Transparent Peer Review file · Nature Communications]

KATANIN promotes cell elongation and division to generate proper cell numbers in maize organs

Corresponding Author: Professor Carolyn Rasmussen

Version 0:

Reviewer comments:

Reviewer #1

(Remarks to the Author)

Martinez et al. reported an interesting study about an evolutionarily conserved microtubule severing protein, KATANIN, which promotes cell elongation and division to generate proper cell numbers in maize organs. Overall, it is novel and important to understand the functional conservation and divergence of MT-related proteins in cell growth. The presented data are solid and of high-quality, mostly supporting the main conclusion. Nonetheless, additional evidences are needed to further improve this study, including genetic analysis, MT assay in root.

Major concerns:

1. The genetical characterization of the *dcd3a* and *dcd3b* mutants are not enough, and remain to be further elucidated. (1) For *dcd3a* mutants, a validation for the mutations in *DCD3a* is necessary, by using either allelic test or functional complementation. Since either single mutant has no phenotype, combinatorial double mutants will help to validate. It seems that *dcd3a-1* is caused by a natural variation, like *CML228*. I think a direct crossing between the original *dcd3* with *CML228* is needed, although the essential mutation site is still unknown. (2) For *dcd3b* mutants, a validation is also necessary, as mentioned for *dcd3a*. (3) For the *Ctl1* mutant, genetically it seems like a semi-dominant mutant, however, functionally it appears to be a weak mutant. Therefore, it still needs some genetic experiments to validate, such as crossing between *Ctl1* and *dcd3b*. (4) The conclusion that the two natural variations of *DCD3a*, *CML228* and *CML333*, are enhancers of *Ctl1* is difficult to understand, if *Ctl1* is a real dominant mutant. Therefore, I suggest more genetic analysis is needed to explain this paradox.
2. As the authors have shown that the genetic backgrounds are pivotal to phenotypical variations, it should be mentioned whether these mutants are somehow near isogenic lines in this study.
3. For cell elongation and division assay, I think root is an ideal system rather than leaf. The microtubule morphology distribution and orientation are much easier to detect. The authors give a more general and stronger conclusion as showing in the title, I strongly suggest the author to verify the conclusion in root system. Importantly, the orientation of microtubule array is important to cell elongation, such as the angles between cortical microtubule and the horizontal axis in root. Moreover, at least one single mutant is needed to use as a control for the MT assay.
4. The authors need to clearly distinguish the cortical and phragmoplast MTs, as they are functionally and organized different. I suggest to separate them to get a more accurate and concise conclusion.
5. A thorough discussion of the KATANIN function conservation and divergence is needed, specifically between maize and *Arabidopsis* and their animal counterparts.
6. A MT drug assay for the mutants could be a plus for the study.

Minor concerns:

1. L35, Abstract the Latin "Zea mays" should be "Zea mays L.", the same for L55
2. Fig1, the mutation details are needed in A; A quantification of epidermal peel division plane orientation defects is needed in D.
3. Fig 2, C the anisotropy could be improved as MT assay orientation (the angle with growth axis), which is indicative to cell growth.
4. Defects in cell division are also needed to show its roles in phragmoplast MT array formation and cell division.

Reviewer #2

(Remarks to the Author)

The discovery, function of katanin, highlighted in this manuscript is fundamentally super-important for basic cell biology. Authors did a fantastic job in quantitative cell biology part. The manuscript is well written and one of the best manuscripts I read in current years. I have a few comments and suggestions for authors to consider and I have mentioned them in the following section.

In Figure 1, what is the frequency of defective division plane in *dcd3a-2 dcd3b* mutant.

In Figure 3, it will be nice to show some representative cell images used for the quantification.

In Figure 5D and F, the uneven and one-sided PPB are interesting. If author has Z-stacks of similar movies/images, it will be interesting to show the 3D reconstruction of uneven and one-sided PPB.

It is interesting that final cell division planes are correct, although the cell starts with a partially formed PPB (L257-L258). Another cool observation is that PPB position influences nuclear position in *dcd3a-2 dcd3b* mutant (L281-L282). Do you think, in a cell type, where nucleus moves polarly in pre-mitotic condition, the nuclear position might be affected differently?

L133: Mention the microtubule marker used.

Reviewer #3

(Remarks to the Author)

Katanin-mediated microtubule severing is critical for proper cell elongation and cell division in plants. Mutants of the p60 catalytic subunit of katanin have been characterized in multiple plant species, including the monocot crop rice. In this manuscript, the authors characterize maize mutants of katanin p60 homologs (the *dcd3a-2 dcd3b* double mutant). The role of katanin in microtubule severing, organization, and division plane positioning has already been extensively characterized in *Arabidopsis* and investigated in several other plant species. Although I appreciate the authors' detailed analysis of the maize mutants and agree that understanding how katanin regulates plant growth and development is an important question worthy of further investigation, I am not fully convinced that this study provides sufficient new insight into the molecular mechanisms by which plant katanin regulates cell division and organ growth. Substantial additional work is needed to elucidate any unique or previously uncharacterized functions of katanin in maize. I also recommend a more careful interpretation of the current data, avoiding over-speculation beyond what the evidence directly supports. My specific concerns are outlined below.

1. Due to the complexity of the genetic background in EMS-generated mutants, further characterization of the *dcd3a* and *dcd3b* alleles is necessary. The nature of the *Clt1* allele also requires clarification. Why does it exhibit a semi-dominant effect? Is this due to haploinsufficiency or a dominant-negative action of the mutated protein? The authors report that the *dcd3a-2 Clt1* double mutant is lethal; however, based on the current data, it remains inconclusive whether complete knockout of both *Dcd3a* and *Dcd3b* results in lethality. Given that *KTN1* is not an essential gene in *Arabidopsis*, generating CRISPR/Cas9 knockout lines for *Dcd3a* and *Dcd3b* would help clarify this issue. The *dcd3a-2 dcd3b* double mutants are reported to be sterile. What is the underlying cause of this sterility? Are *Dcd3a* and *Dcd3b* required for male and/or female gametophyte development? The authors didn't observe decreased severing in *Clt1/+*. It is possible that the developmental defects seen in *Clt1/+* are influenced by additional, uncharacterized mutations in this maize strain. Molecular complementation of the *Clt1* and *dcd3a dcd3b* double mutants with wild-type *Dcd3a* or *Dcd3b* expression constructs would be important to confirm that the observed phenotypes are indeed caused by mutations in these genes.
2. What is the subcellular localization of maize *Dcd3a* and *Dcd3b* proteins? Do they localize to MT crossovers, similar to katanin in other species? It would also be informative to examine the localization patterns of *Dcd3a* and *Dcd3b* at different stages of mitosis to better understand their potential roles in cell division.
3. The authors attempt to link the reduced organ size and decreased cell division rate observed in the *dcd3a-2 dcd3b* mutants. However, the current evidence does not distinguish whether the smaller cell size is associated with a delay in the G1 or G2 phase of the cell cycle. Alternatively, the reduced number of dividing cells could simply reflect that fewer cells in the *dcd3a-2 dcd3b* mutants are competent to divide.
4. Is there any defects in phragmoplasts in *dcd3a-2 dcd3b*? Current data cannot rule out that defects in phragmoplast may contribute to mispositioned cell walls.
5. Based on the authors' data, it is also possible that the mispositioned nucleus contributes to the altered positioning of the PPB. It would be helpful for the authors to clarify whether katanin directly regulates nuclear positioning, or if the observed mispositioned nucleus is a downstream effect of PPB defects. Addressing this distinction would strengthen the mechanistic interpretation of the data. The authors may wish to discuss the significance of nuclear positioning in regulating the division plane and its potential impact on cell elongation.
6. In the Discussion, the authors suggest that the PPB may play a more important role in division plane positioning during asymmetric divisions. However, such analyses are not performed in the current manuscript.
7. The authors report that MT severing frequency was reduced, but not completely abolished, in the *dcd3a-2 dcd3b* mutants.

While several potential explanations are discussed, additional experiments would be necessary to distinguish between these possibilities.

Minor points:

1. For clarity and consistency, the *dcd3b* allele used in the *dcd3a-2 dcd3b* combination should be designated as *dcd3b-1*, which would also facilitate systematic naming of any additional *dcd3b* alleles identified in future studies.
2. In Fig. 1A, a clearer graphical representation is needed to more effectively convey the information about the *dcd3a* and *dcd3b* alleles analyzed in this study.
3. In Fig. 5C, additional labels are necessary to clearly indicate the types of cells that were measured.
4. Time-lapse imaging videos should be provided.

Version 1:

Reviewer comments:

Reviewer #1

(Remarks to the Author)

The authors almost properly addressed all my concerns, and the quality of this revision is greatly improved. I have some minor points that possibly could further strengthen the conclusion of this study.

1. The explanation of Ctl1 phenotype could be validated by the checking the direct interactions between *Dcd3a* and *Dcd3b*, either heterodimer or homodimer. This is also important to explain the phenotypes of Ctl1 enhancers.
2. It will be good to show the *Dcd3b* sequences in those enhancer inbreds. Is there any explanation for *DCD3* variations, especially the loss-of-function of *Dcd3a*, during maize evolution.
3. There are eight amino acid substitutions in CML228. I suggest the authors narrow down the numbers.
4. I think the Abstract could be improved to address the major findings in this study.
5. the Latin "Zea mays" should be "Zea mays L.", L should not be italic.

Reviewer #2

(Remarks to the Author)

Thanks for going through each comments and improve the figures and add explanation. I really appreciate it. It's an excellent research article. Looking forward to seeing the article in print. Congratulations!

Reviewer #3

(Remarks to the Author)

I appreciate the authors' thorough responses to my questions. The manuscript has been substantially improved by the additional work, and I have no further major concerns.

Minor point:

In the legend of Figure 4A, "YFP-TUBULIN (green) and TAN1-YFP" may need to be CFP-TUBULIN rather than YFP-TUBULIN.
